# Establishing microbial composition measurement standards with reference frames

James T. Morton[1,2,10], Clarisse Marotz [1,10], Alex Washburne[3], Justin Silverman[4,5,6], Livia S. Zaramela [1], Anna Edlund[7], Karsten Zengler [1,8,9,11] & Rob Knight [1,2,9,11]

Differential abundance analysis is controversial throughout microbiome research. Gold standard approaches require laborious measurements of total microbial load, or absolute number of microorganisms, to accurately determine taxonomic shifts. Therefore, most studies rely on relative abundance data. Here, we demonstrate common pitfalls in comparing relative abundance across samples and identify two solutions that reveal microbial changes without the need to estimate total microbial load. We define the notion of "reference frames", which provide deep intuition about the compositional nature of microbiome data. In an oral time series experiment, reference frames alleviate false positives and produce consistent results on both raw and cell-count normalized data. Furthermore, reference frames identify consistent, differentially abundant microbes previously undetected in two independent published datasets from subjects with atopic dermatitis. These methods allow reassessment of published relative abundance data to reveal reproducible microbial changes from standard sequencing output without the need for new assays.

[1] Department of Pediatrics, University of California, San Diego, La Jolla, CA 92093, USA. [2] Department of Computer Science & Engineering, University of California, San Diego, La Jolla, CA 92093, USA. [3] Department of Microbiology and Immunology, Montana State University, Bozeman, MT 59717, USA. [4] Program in Computational Biology and Bioinformatics, Duke University, Durham 27708, USA. [5] Medical Scientist Training Program, Duke University, Durham 27708, USA. [6] Center for Genomic and Computational Biology, Duke University, Durham 27708, USA. [7] J. Craig Venter Institute, Genomic Medicine Group, La Jolla, CA 92037, USA. [8] Department of Bioengineering, University of California, San Diego, La Jolla, CA 92093, USA. [9] Center for Microbiome Innovation, University of California, San Diego, La Jolla, CA 92093, USA. [10] These authors contributed equally: James T. Morton, Clarisse Marotz [11] These authors jointly supervised this work: Karsten Zengler, Rob Knight Correspondence and requests for materials should be addressed to K.Z. (email: kzengler@ucsd.edu) or to R.K. (email: rknight@ucsd.edu)

Next-generation sequencing data used to study the microbiome is inherently compositional and provides information in the form of relative abundances, independent of the total microbial load of the original sample. Numerous analytical approaches including rarefaction[1], median[2], and quantile normalization[2,3] have been proposed for comparing compositional samples. However, these analytical solutions cannot control false discovery rates[4,5], and their application contributes to lack of reproducibility among microbiome studies[6–8]. Here we illustrate mathematical challenges in analyzing compositional microbiome data from DNA sequence reads, and define the concept of "reference frames" for inferring changes in abundance.

To illustrate the pitfalls of inferring changes in abundance among samples using relative abundance data, consider the following example (Fig. 1). Samples from a population containing only two taxa (orange and blue) are collected pre- and post-treatment. Before treatment, the two taxa occur in equal proportions. After treatment, the orange taxon is twice as abundant as the blue taxon. It is tempting to conclude that orange increased and blue decreased.

However, many different scenarios could lead to the same observation. For example, the orange taxon could quadruple and the blue taxon only double. The orange taxon could remain constant, and the blue taxon halve. Or the orange taxon could halve, but the blue taxon could decrease four-fold. Because we only observe relative abundance data, we cannot differentiate among these outcomes, which have markedly different biological significance. Infinite different outcomes produce the same 2:1 ratio of orange to blue, greatly complicating the generation of a meaningful null hypothesis and therefore yielding misleading $p$-values, as has been previously established[9–11].

Multiple processing steps are required to generate microbiome sequencing data. Samples are collected from a much larger population (e.g., fecal material from the gut, or water sample from the ocean). From these samples, a subsample is used for DNA extraction (e.g., a swab from a fecal sample, or an aliquot of a water sample). Even if the same amount of sample is extracted throughout an experiment, many DNA extraction kits are optimized for efficiency and can become saturated, complicating direct correlations of DNA yield and microbial load. A subsample of the extracted DNA is then used as input for PCR, a subset of the resulting amplicon is pooled into a library, and a subset of the library is sequenced.

By the time quality-filtered sequencing data are obtained, the sequences reflect only a small subset of the population and are not an accurate representation of the microbial load in the original sample[12]. Analyzing relative abundance data with inappropriate statistical tools can yield up to 100% false discovery rates[13,14]. Therefore, in addition to relative abundance data, quantitative information about total microbial load is necessary to determine which microbes are changing.

Multiple approaches at each level of sample processing have been proposed to quantify the total microbial load from environmental samples. Adding a known amount of reference DNA as an internal standard has been used to extrapolate the amount of starting nucleic material[15,16]. Normalization by this method is complicated due to the calibration challenges of choosing the proper amount of internal standard[16]. At the post-extraction level, quantitative PCR (qPCR) of genomic DNA with universal primers against the 16S rRNA gene has been deployed to estimate total microbial load[17]. However, it is impossible to prevent primer bias, resulting in uneven amplification of rRNA genes across species, and the DNA extraction method can influence microbial composition[18–20]. Further, quantification by both spike-in and qPCR is performed on multiple subsets of the original sample.

Quantifying microbial load by flow cytometry is performed on the original sample, and is agnostic to nucleotide sequences. One recent study reported that adding quantitative information obtained by flow cytometry dramatically improved interpretation of 16S rRNA gene amplicon sequencing data[12]. However, flow cytometry requires expensive, relatively low-throughput equipment, and often can only estimate the cell concentration rather than the total microbial load.

The total microbial load of an environmental sample is one dimension of measurement among the hundreds to thousands of dimensions measured by microbial relative abundances. If the absolute abundance of one taxon and the relative abundance of all taxa is known, it is feasible to compute the absolute abundance of all taxa. As such, considerable information rests in relative abundances, and important insights can be gleaned without costly microbial quantification methods. Below we describe two methods to evaluate relative differential abundance independent of microbial load information.

## Results

### Ratios circumvent bias without microbial load quantification.

Computing changes in abundance from compositional data introduces a bias due to the lack of total microbial load (Fig. 1 approach#1). Simulated data in Fig. 1b shows how different biases (i.e., ratios between total microbial loads) can cause either false-positives or false-negatives. By simply comparing the ratio of taxa between samples, the bias constant introduced by unknown microbial load cancels out. Taking the logarithm of this ratio (log-ratio) enforces symmetry around zero, giving equal weight to relative increases and relative decreases[9,10].

### A novel approach to rank differential abundance.

Comparing ratios of taxa can circumvent the bias introduced by unknown microbial loads. However, choosing taxa for comparison from the thousands in a given sample set can be challenging. Here we provide a way to rank microbes that are changing the most relative to each other. The term "differential" refers to the logarithm of the fold change in abundance of a taxa between two conditions. With microbial load information, one can calculate absolute differentials. Microbiome sequencing datasets provide relative abundances, and thus can only infer relative differentials.

The ranks of relative differentials are identical to the ranks of absolute differentials (Fig. 1d). However, because of the bias described above, we cannot infer if a microbe has changed based on rank alone, and therefore a coefficient of zero does not imply that the microbe has not changed abundance.

Relative differentials can be estimated directly using multinomial regression, which has been proposed previously to handle sampling zeros[21–24]. The coefficients from multinomial regression analysis can be ranked to determine which taxa are changing the most between samples. We refer to this ranking procedure as differential ranking (DR).

### Reference frames in compositional data analysis.

Analyzing compositional data requires a choice of reference frames for inferring changes in abundance. By "reference frame", we draw on the concept from physics where velocity is measured relative to another moving object. As microbial populations change, we can constrain our inferences to how microbial populations change relative to reference frames given by other microbial populations. The denominator in a log-ratio determines the reference frame for inferring changes. In DR, the differential abundance of each taxon serve as a reference to each other when they are ranked numerically. To demonstrate these principles, we

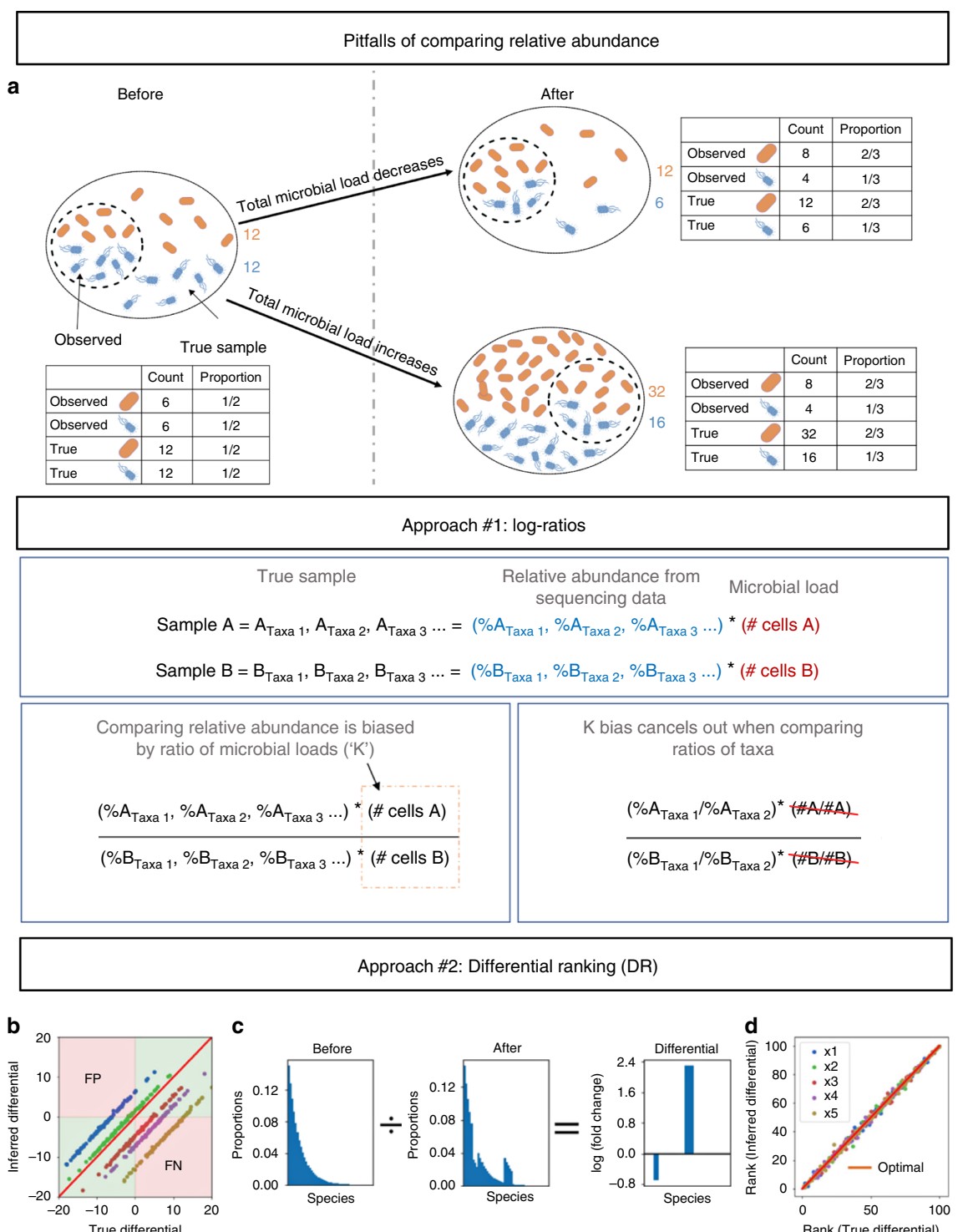

**Fig. 1** Illustration demonstrating statistical limitations inherent in compositional datasets. **a** Two different biological scenarios can yield the exact same proportions of taxa in samples from a population pre- and post-treatment. **b** Simulated datasets plotting the true differential obtained using absolute abundance data on the *x*-axis, versus the inferred differential obtained using relative abundance data on the *y*-axis. Each dot represents a taxon in the dataset, and the colors represent datasets with various ratios of total microbial load (K) between before and after samples. The red line represents the optimal scenario where the samples have equal microbial load. This illustrates the prevalence of either false positives (FP) or false negatives (FN) when performing differential abundance analysis on samples with unequal total microbial load. The presence of either FPs or FNs is dictated by a nonlinear function of the true differential (see online methods). **c** An illustration of differential proportions of bacterial species before and after treatment. **d** Same data as **b** but plotting the rank of the differentials, demonstrating that ranks are equivalent regardless of differences in microbial load

confirm the utility of employing reference frames in biological datasets.

**DR reveals differentially abundant microbes in saliva.** We demonstrate the utility of DR in a sample set with dramatic differences in total microbial load. Unstimulated saliva samples were collected before and after brushing teeth ($n = 32$), and processed in parallel for microbial load quantification with flow cytometry and 16S rRNA gene amplicon sequencing. Importantly, participants were asked to provide unstimulated saliva for exactly 5 min. As a result, we obtained a proxy for the total microbial load by taking into account salivary flow rate. As expected, the total microbial load significantly decreased after brushing teeth (Fig. 2a).

We performed paired $t$-tests to evaluate the change in abundance of each taxon before and after brushing teeth using either relative or absolute abundance data (microbial load multiplied by 16S copy number-corrected relative abundances) (Fig. 2c). Applying $t$-tests to the relative data had a high false-positive rate, as seen by the disagreements between the relative and absolute $t$-statistics (Spearman $r = 0.53$). Further, there was no correlation in $p$-value distribution between the relative and absolute abundance data (Spearman $r = 0.09$), highlighting issues when the null hypothesis is not consistent between the relative abundances and the absolute abundances.

Alternatively, evaluating the ratio between *Actinomyces* and the remaining taxa produced identical $t$-statistics and $p$-values between the relative and absolute abundance data (Spearman $r = 1.0$). Ratio-based analyses are unaffected by microbial load (Eq. (3) in methods) and result in identical interpretations as one obtains from costly and rate-limiting flow-cytometry measurements.

From the DR analysis (Fig. 2c), we can identify which taxa are changing the most relative to each other. Here, we highlight *Actinomyces* and *Haemophilus* species, which have very different ranks. *Actinomyces* tend to have low ranks and *Haemophilus* have high ranks. The difference in ranks between these taxa correctly suggests that *Haemophilus* taxa are more prevalent relative to other taxa before brushing, and *Actinomyces* taxa are more prevalent relative to other taxa after brushing. From the $t$-test results on relative abundances it appears that *Actinomyces* significantly increased ($t$-statistic = 3.74, $p$-value = 0.002) after brushing teeth and that *Haemophilus* significantly decreased ($t$-statistic = −3.67, $p$-value = 0.002). However, absolute abundance data revealed that only *Haemophilus* significantly decreased ($t$-statistic = −2.155, $p$-value = 0.0478) (Fig. 2d).

The log-ratio of *Actinomyces* and *Haemophilus* between the relative and the absolute abundance data is identical. While we cannot observe the decrease of *Haemophilus* or the consistency of *Actinomyces* abundance, with the log-ratio of their relative abundance we can observe the interaction between these two taxa and the increase of *Actinomyces* relative to *Haemophilus* after brushing teeth ($t$-statistic = 5.289, $p$-value = $9.07 \times 10^{-5}$).

These results are consistent with our knowledge about oral biogeography. *Haemophilus* is typically found on the periphery of oral biofilms and was likely removed from the biofilm during the brushing process, whereas *Actinomyces* is generally found on the surface of the tooth and acts as an anchor for biofilm attachment[25]. Importantly, this experiment demonstrates the potential fallibility of relying on relative abundance; it is incorrect to conclude that *Actinomyces* increases after tooth brushing despite the increase in relative abundance. As demonstrated by flow cytometry, total microbial load decreases, and while both *Haemophilus* and *Actinomyces* decrease, *Haemophilus* decreases more.

To investigate how other compositional methods perform, we ran ANCOM and ALDEx2 on the same dataset (Fig. 2e). ALDEx2 did not identify any of the microbes to be changing, which contradicts flow-cytometry measurements that show there is a large decrease in the microbial community after tooth brushing. ANCOM identified multiple significantly changing microbes. One of these detected microbes was *Veillonella*, which conflicts with absolute abundances suggesting that *Veillonella* is not significantly changing ($t$-statistic = 1.04, $p$-value = 0.315). The false-positive detected by ANCOM likely arose due to their choice of reference frame (Supplementary Fig. 1).

**Elucidating interkingdom changes in atopic dermatitis using DR.** The tooth brushing example provides ground truth for using log-ratios and DR, but many clinically relevant microbiome questions involve less obvious differences. Using data from patients with atopic dermatitis (AD), an important skin disease, we demonstrate how viewing relative abundances alone can produce false negatives.

AD has a complex etiology. Many microbiome studies performed using next-generation sequencing have focused on bacterial changes associated with AD, especially the pathogen *Staphylococcus aureus*. The yeast genus *Malassezia* has also been implicated in AD, although conflicting results have been published as to which *Malassezia* species are involved and whether they are more or less prevalent in AD[26]. A recent shotgun metagenomic study examined the skin microbiome over time during an AD flare and recovery. The authors observed a decrease in *Staphylococcus aureus* relative abundance in the healthy, recovered skin (non-lesioned) compared to AD flare (lesion), but no significant changes in the relative abundance of *Malassezia* species over time in these AD patients[27].

Applying compositonal methods to this dataset revealed new insights. Observing the DR results (Fig. 3a), it is apparent that, compared to lesioned skin, *S. aureus* is one of the taxa to decrease the most relative to all other microbes in the non-lesioned sites, followed by *S. epidermidis*, and *M. globosa*. Consistent with the analysis of relative abundance in Fig. 3b, the ratio of *S. aureus*: *P. acnes* was significantly increased in flare ($t$-statistic = 3.397, $p$-value = $3.02 \times 10^{-3}$) and correlated with SCORAD score, a clinical assessment of AD severity (Pearson = 0.603, $p$-value = $3.516 \times 10^{-6}$). Contrary to previous findings, both *S. epidermidis*: *P. acnes* and *M. globosa*: *P. acnes* were also significantly increased in lesioned skin ($t$-statistic = 4.2297, $p$-value = $4.53 \times 10^{-4}$, and $t$-statistic = 4.297, $p$-value = $3.889 \times 10^{-4}$, respectively) and correlated with SCORAD score (Pearson $r = 0.464$, $p$-value = $6.975 \times 10^{-4}$, and Pearson $r = 0.668$, $p$-value = $1.125 \times 10^{-7}$, respectively) (Fig. 3c).

To validate this observation, we analyzed shotgun data from an independent AD dataset[28]. In this dataset, the relative abundance of *M. globosa* significantly increased between lesioned and non-lesioned skin (Fig. 3e, $t$-statistic = 4.135, $p$-value = 0.0001). But the ratio of *M. globosa*: *P. acnes* increased even more dramatically in lesioned skin ($t$-statistic = 7.298, $p$-value = $9.729 \times 10^{-9}$) (Fig. 3d). These results are congruent with a previous report that *M. globosa* was cultivated more successfully from lesioned versus non-lesioned sites in AD[29]. Thus, DR analysis can identify novel, clinically significant microbial changes, which can be validated across cohorts by choosing insightful reference frames.

**DR across environmental gradients in the Central Park soils.** Differential ranks can also be learned for continuously valued data. We demonstrate this with data from the Central Park soil experiment[30], which contains more than 1000 samples and

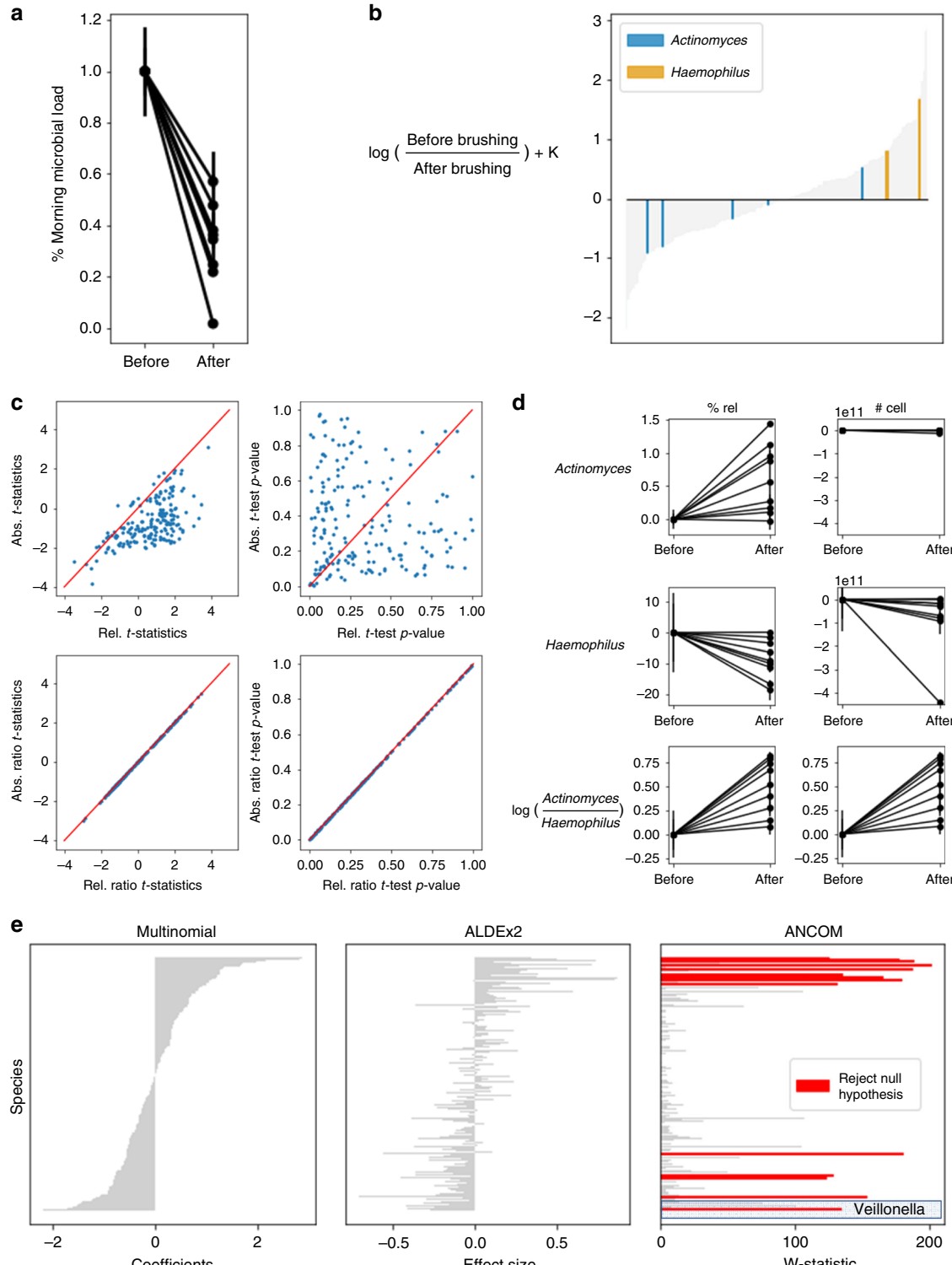

**Fig. 2** Analysis of salivary microbiota before and after brushing teeth. **a** Flow-cytometry-quantified microbial load in unstimulated saliva collected for 5 min normalized to before brushing teeth. Each line corresponds to a different volunteer. Error bars represent the standard deviation from duplicate flow-cytometry measurements. **b** Microbial ranks estimated from multinomial regression with *Actinomyces* and *Haemophilus* highlighted. The y-axis represents the log-fold change that is known up to some bias constant K, and the x-axis numerically orders the ranks of each taxa in the analysis. **c** A comparison of t-statistics (left) and p-values (right) between before and after samples where each dot is an individual taxon (top graphs) or ratio between each taxon to *Actinomyces* (bottom graphs) calculated from relative abundance data (x-axis) and absolute abundance data (y-axis). The 1-1 correspondence in the ratio graphs is a result of the microbial loads cancelling out, as described in Eq. (3). **d** A comparison of relative abundance vs absolute abundance data of *Actinomyces*, *Haemophilus* and log(*Actinomyces: Haemophilus*) before and after brushing teeth. Error bars represent standard error of the mean. **e** Comparison of the multinomial coefficients used for DR, ALDEx2 and ANCOM outputs. The test statistics generated from ALDEx2 and ANCOM are sorted in the same order as the multinomial coefficients to provide a consistent comparison. All taxa that passed the significance tests are highlighted in red

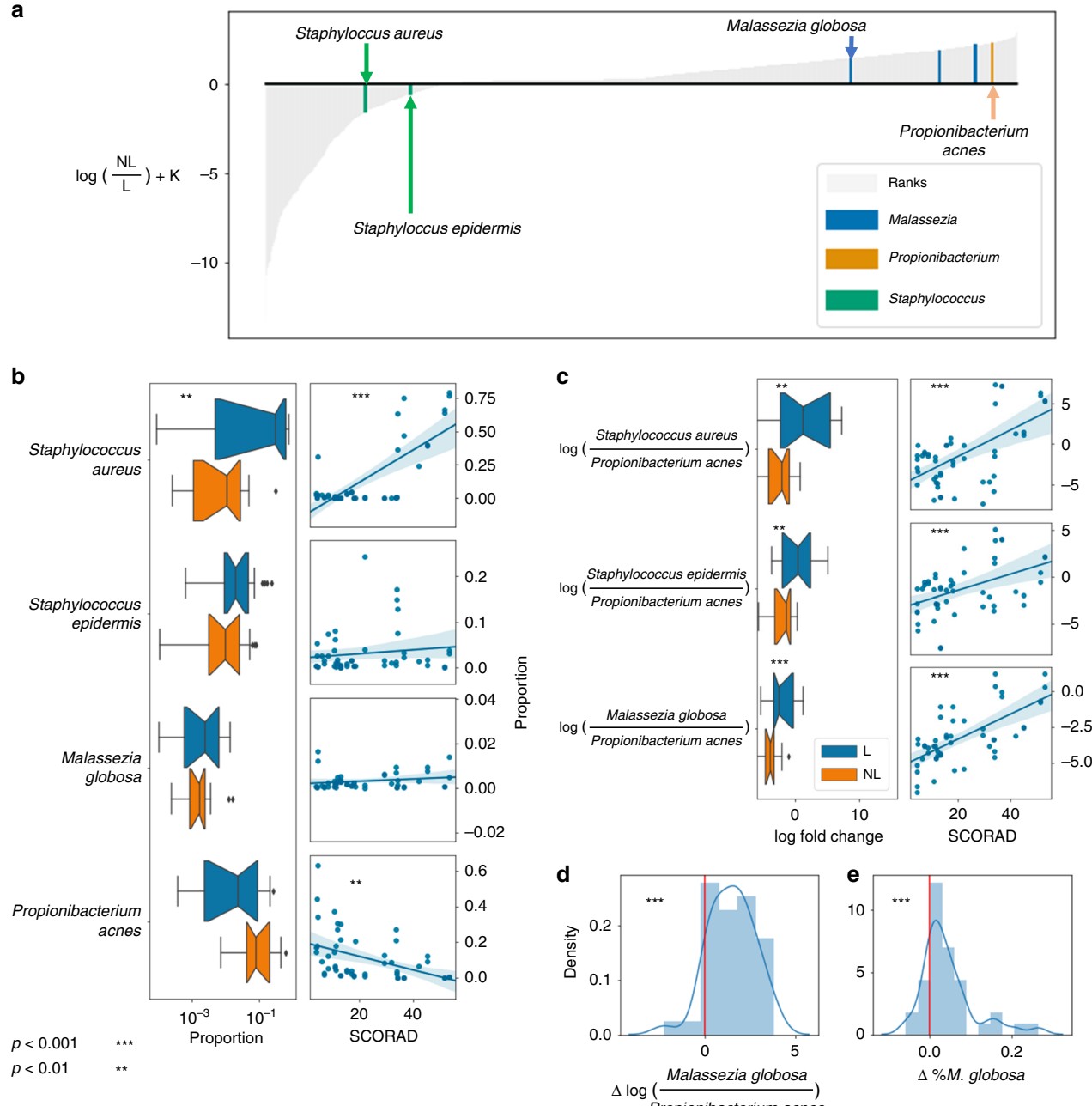

**Fig. 3** DR analysis of skin in two atopic dermatitis studies. Panels **a–c** represent data from Byrd et al.[27], and panels **d**, **e** represent data from Leung et al.[28]. Both studies compare lesioned (L) to non-lesioned (NL) skin. **a** Microbial ranks estimated from multinomial regression applied to shotgun metagenomics from Byrd et al.[27] with key genera highlighted. The y-axis represents the log-fold change that is known up to some bias constant K. **b** Proportions of *S. aureus*, *S. epidermidis*, *M. globosa*, and *P. acnes* in lesioned (blue) and non-lesioned (orange) skin (left) and correlation of relative abundance with SCORAD score (right). **c** Log-ratios of (*S. aureus: P. acnes*), (*S. epidermidis: P. acnes*), and (*M. globosa: P. acnes*) (left) and correlation of ratio with SCORAD score (right). Error bars represent standard deviation across participants (n = 20). **d** Change in log-ratio of (*M. globosa: P. acnes*) from Leung et al.[28]. **e** Change in relative abundance of *M. globosa* between lesioned and non-lesioned skin from Leung et al.[28]. Presented p-values are from paired t-test statistics

30,000 taxa sampled across pH and nitrogen gradients. The largest factor driving diversity was pH, and Washburne et al.[31] showed that there were lineages of microbes associated with nitrogen when pH was accounted for. Here we applied multinomial linear regression to estimate microbial DR along both nitrogen and pH gradients (Fig. 4).

Of the top five and bottom five ranked microbes in the nitrogen and pH gradients, only four microbes were annotated. The top fourth and fifth microbe that is associated with acidic environments was a putative match against *Candidatus Solibacter*

and *Telmatobacter*, which has been found to grow in a pH range of 3.5–6[32,33]. The top microbe most associated with high pH was *Chryseolinea*, which has been shown to grow between pH range of 5–10. The top third microbe associated with low nitrogen concentration was a putative match against *Gemmatimonas*, which is a known nitrogen reducer[34].

The multinomial regression was able to appropriately identify which organisms were most associated with low pH, high pH, and nitrogen. However, even amongst the highly ranked organisms there is a major lack of functional annotations. Having the

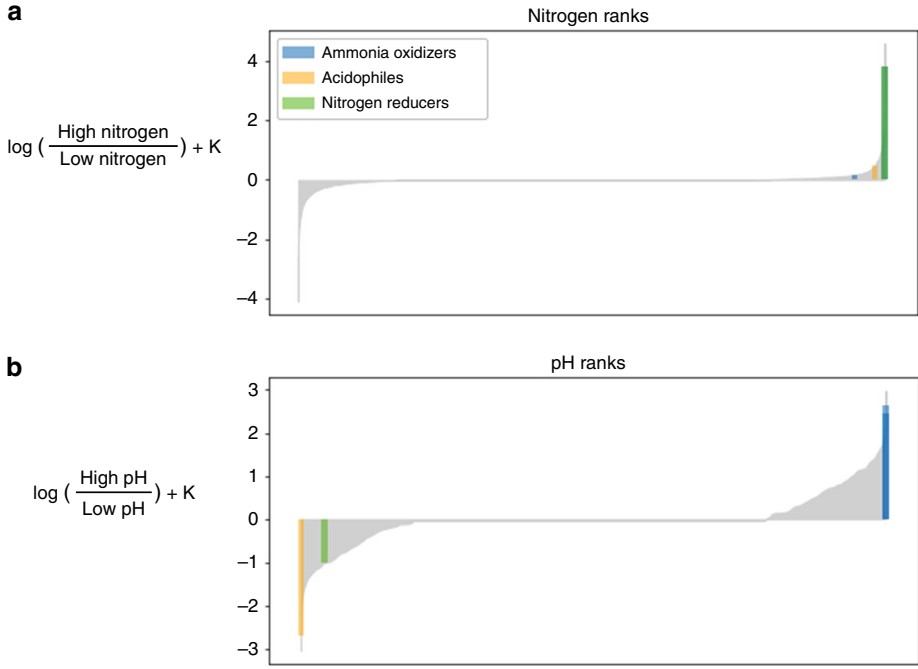

**Fig. 4** DR analysis of the Central Park dataset. **a** Microbes ranked with respect to their association with nitrogen. **b** Microbes ranked with respect to their association with pH. Putative hits against an acidophile, an ammonia oxidizer and a nitrogen reducer are highlighted

appropriate rankings in place may provide new insights into these organisms and guide experimental validation.

## Discussion

Adding information about absolute microbial load between samples can highlight issues inherent in compositional data analysis. However, there are multiple practical and technical challenges in quantifying microbial load. For example, skin swabs are often difficult to use in flow cytometry due to very low microbial load and difficulty in transferring intact cells from swabs into liquid solution. Furthermore, skin samples are notoriously sensitive to 16S rRNA gene primer choice making qPCR quantification challenging[35]. Similarly, for historically collected samples that exist only as DNA in a freezer or as sequences in a database, flow cytometry approaches to determine absolute microbial load are not feasible.

However, absolute abundances of a community are only one degree of freedom; in a community of N species, N − 1 degrees of freedom exist in the relative abundances. By using flow cytometry to quantify total microbial load, we validated these analytical tools in 16S rRNA gene amplicon sequencing data from unstimulated saliva. We found evidence of false-positives when looking exclusively at changes in relative abundance before and after brushing teeth. By evaluating the ratio of *Actinomyces: Haemophilus*, we reached an identical conclusion to our absolute abundance data without the need for microbial load quantification. The consistency of our results rests in the use of ratios defining reference frames for inferring compositional changes.

Furthermore, we highlighted an example of a false-negative in previously generated shotgun metagenomic data from the skin of individuals with AD. We were able to reproduce the findings that *S. aureus*, and to a lesser extent *S. epidermidis*, are differentially abundant in AD lesions. Additionally, using log-ratios and differential ranking, we were also able to show a more subtle but statistically significant change in *M. globosa* abundance in AD lesions. This same result was obtained in two independent metagenomic studies of AD patients and agrees with previous cultivation-based work quantifying increased colony forming units of *M. globosa* in AD lesions.

Consistency between inferences made based on relative and absolute abundance data is crucial, because in many circumstances it is not possible or practical to estimate total microbial load. The seeming contradiction between absolute and relative abundances does not invalidate data from the existing 100,000+ experiments utilizing 16S rRNA gene amplicon or metagenomic sequencing[36,37]. Importantly, these techniques are not limited to next-generation microbiome sequencing, but can be applied to any experiments involving compositional data (e.g., metabolomics, proteomics, etc.).

Although various methods of multinomial-based models have been developed[21–24], the interpretation of the resulting model requires care. A zero valued coefficient does not imply that the corresponding species abundance has not changed, due to the total microbial load bias as discussed in Fig. 1. DR provides a novel means to correctly interpret the coefficients of these models. By ranking the coefficients we can determine which taxa have changed the most relative to each other. This subtle distinction acknowledges the limits of compositional data analysis, and as demonstrated above can have dramatic impacts on data interpretation.

While there are widespread misconceptions concerning how to interpret microbial abundances, we have shown that misinterpretations stem from a misunderstanding of the reference frame used in analysis. Ongoing efforts at the NIH and EMBL-EBI have already stored petabytes of multi-omics datasets ready to be re-analyzed, and databases, such as Qiita and gcMeta, contain curated data and metadata from hundreds of thousands of samples[36,37]. There is much promise for resolving outstanding controversies by re-analyzing these datasets using reference frames to make stable inferences of compositional change.

## Methods

**Cancelling out bias in compositional data**. The change between two samples containing compositions (e.g., absolute abundances of $D$ microbes) $\boldsymbol{A} = (a_1, ..., a_D)$

and $\boldsymbol{B} = (b_1, \ldots, b_D)$, can be computed as follows

$$\frac{\boldsymbol{A}}{\boldsymbol{B}} = \left( \frac{a_1}{b_1}, \ldots \frac{a_D}{b_D} \right) \tag{1}$$

If we are only able to measure relative abundances, as is the case with next-generation amplicon sequencing, we can only estimate the proportion $p_{a_i}$ for species $i$ in the sample $A$ (i.e., $p_{a_i} = \frac{a_i}{N_a}$). Estimating the true abundance can be done via $a_1 = N_a p_{a_1}$, where $N_a$ is the total abundance of sample $A$. To estimate the true change,

$$\frac{\boldsymbol{A}}{\boldsymbol{B}} = \left( \frac{N_a \times p_{a_1}}{N_b \times p_{b_1}}, \ldots \frac{N_A \times p_{a_D}}{N_B \times p_{b_D}} \right) = \frac{\boldsymbol{p_A}}{\boldsymbol{p_B}} \times \frac{N_A}{N_B} \tag{2}$$

To determine if species $i$ abundance has changed between samples $A$ and $B$, we test to see if $\frac{a_i}{b_i} = 1$. However, as shown above, we cannot perform this test, since the results of this test would be confounded by the total biomass bias $\frac{N_A}{N_B}$.

In many cases the total biomass cannot be estimated, so any techniques to identify important species will need to alleviate this bias. One alternative is to use ratios. If we choose species D to be the reference species, it is clear that the total biomass cancels as follows

$$\frac{a_1/a_D}{b_1/b_D} = \frac{p_{a_1}/p_{a_D}}{p_{b_1}/p_{b_D}} \tag{3}$$

Another alternative is to use ranks. Ranks have been shown to be context of microbiome studies[38,39] and have been commonly employed to study species richness in the context of ecology[40]. Since the bias is applied uniformly across the differential, it will not affect the ordering of the species. Hence, ranks are agnostic to the total biomass bias.

$$\mathrm{rank}\left(\frac{\boldsymbol{A}}{\boldsymbol{B}}\right) = \mathrm{rank}\left(\frac{\boldsymbol{p_A}}{\boldsymbol{p_B}} \times \frac{N_A}{N_B}\right) = \mathrm{rank}\left(\frac{\boldsymbol{p_A}}{\boldsymbol{p_B}}\right) \tag{4}$$

Because of the equivalence of ranks between absolute and relative data, it is possible to identify the species that are increasing or decreasing the most. This means that the following statements hold

$$\arg\max\left(\frac{\boldsymbol{A}}{\boldsymbol{B}}\right) = \arg\max\left(\frac{\boldsymbol{p_A}}{\boldsymbol{p_B}}\right) \tag{5}$$

$$\arg\min\left(\frac{\boldsymbol{A}}{\boldsymbol{B}}\right) = \arg\min\left(\frac{\boldsymbol{p_A}}{\boldsymbol{p_B}}\right) \tag{6}$$

The ranks are connected to the log-ratios, the differences between ranks will yield differences in log-ratios given by

$$\log\frac{a_i/N_a}{b_i/N_b} - \log\frac{a_j/N_a}{b_j/N_b} = \log\frac{a_i/b_i}{a_j/b_j} = \log\frac{a_i/a_j}{b_i/b_j} \tag{7}$$

These ranks are still relative; a microbe that is detected to be increasing the most could still be decreasing in absolute abundance. For instance, in the tooth brushing example, ranks identified specific genera of *Actinomyces* to be increasing the fastest, but all of the microbes are depleted, suggesting that *Actinomyces* is just decreasing much less than the other microbes.

This differential is also commonly referred to as a perturbation in the context of the compositional literature[10]. It is important to note that this does not justify applying rank-based statistical methods, such as Spearman correlation or Kruskal–Wallis, to relative abundance data since these tests do not satisfy scale invariance[41,42].

Both the log-ratios and the differential ranking techniques satisfy scale invariance, meaning that both of these techniques are agnostic to the total microbial load. This concept is critical when analyzing relative abundance data, since this is one step closer to maintaining consistent conclusions between the original environment and the observed sequences.

**False discovery rates in relative differential abundance.** Attempting to estimate absolute log-fold differentials from relative abundances can result in either false-positives (FP) or false-negatives (FN) depending on the distribution of true differential abundance. Whether FNs or FPs are observed depending on a nonlinear relationship involving the true (unobserved) differential abundance. To demonstrate this, let $\boldsymbol{\delta} = (\delta_1, \ldots, \delta_D) = \left(\log\left(\frac{a_1}{b_1}\right), \ldots, \log\left(\frac{a_D}{b_D}\right)\right)$ denote the absolute differential of the $D$ species between two conditions, $A$ and $B$. Further, let $\hat{\boldsymbol{\delta}} = (\hat{\delta}_1, \ldots, \hat{\delta}_D) = \left(\log\left(\frac{a_1/N_A}{b_1/N_B}\right), \ldots, \log\left(\frac{a_D/N_A}{b_D/N_B}\right)\right)$ represent the relative differentials from compositional data. By definition, we know the following is true

$$\log\left(\frac{a_i}{b_i}\right) = \log\left(\frac{a_i/N_A}{b_i/N_B}\right) + \log\left(\frac{N_A}{N_B}\right) \tag{8}$$

If $\log\left(\frac{N_A}{N_B}\right) > 0$, then that will mean that for every microbe $i$, $\delta_i > \hat{\delta}_i$. This implies that there is increased microbial load in $A$ compared to $\boldsymbol{B}$, and that this increase will give rise to FNs. This is because the overall community increase will not be captured from the relative abundance data.

In contrast, if $\log\left(\frac{N_A}{N_B}\right) < 0$, then for every microbe $i$, $\delta_i < \hat{\delta}_i$. This means that there is a decrease in the absolute microbial load in $A$ compared to $B$. This decline in the total community will not be captured from the relative abundances, and some of the species will be detected to be increasing, giving rise to FPs. An example of this was shown in the saliva microbiota study (Fig. 2).

**Multinomial regression.** To perform the differential ranking (DR) analysis, we used multinomial regression. Multinomial regression and related count regression models are commonly used in the context of microbiome analysis. Here, we use the multinomial regression model since these models can reliably estimate the means and can be easily reinterpreted in the context of compositional data analysis.

Counts from the multinomial regression can be formulated using additive log-ratio transformation (alr) in the following generative model

$$\beta_{jk} \sim \mathcal{N}(0, \mu_\beta) \tag{9}$$

$$\boldsymbol{\eta_i} = \mathrm{alr}^{-1}(\boldsymbol{X_i \beta}) \tag{10}$$

$$\boldsymbol{Y_i} \sim \mathrm{Multinomial}(\boldsymbol{\eta_i}), \tag{11}$$

where $\boldsymbol{Y_i}$ represents the measured microbial load for sample $i$. $\boldsymbol{\beta}$ represents the coefficients of the model across all measured covariates indexed by $k$. These coefficients can be interpreted as a relative differential discussed in the examples above. $\boldsymbol{X_i}$ represents the vector metadata covariates for sample $i$. These metadata covariates can represent both continuous and categorical variables, where categorical variables are represented as binary variables. A normal prior centered around zero was placed on the coefficients $\boldsymbol{\beta}$ to serve as regularization to combat issues associated with high dimensionality. The $j$th component of the $\boldsymbol{\beta_k}$ coefficient vector represents the $j$th alr coordinate, which can be interpreted as a log concentration using one of the microbes as a reference. It does not matter which microbe is used as a reference, since the proportions $\boldsymbol{\eta_i}$ will be identical regardless of reference microbe.

The inverse alr function is commonly used in the context of compositional data, given as follows

$$\mathrm{alr}^{-1}(x) = C[\exp(0, x_1, \ldots, x_{D-1})] \quad C[x] = \left[ \frac{x_1}{\sum\limits_{i=1}^{D} x_i}, \ldots, \frac{x_D}{\sum\limits_{i=1}^{D} x_i} \right]. \tag{12}$$

This is also referred to as a degenerate softmax function, which is commonly used in the context of neural networks. This function is isomorphic between $\mathbb{R}^{D-1}$ and $\mathcal{S}^D$ (the space of proportions), so this will ward against identifiability issues when estimating these model parameters. The alr function is defined as

$$\mathrm{alr}(x) = \left( \log\frac{x_2}{x_1}, \log\frac{x_3}{x_1}, \ldots, \log\frac{x_D}{x_1} \right) \tag{13}$$

The models were estimated using a maximum a posteriori priori (MAP) estimation using stochastic gradient descent.

Multinomial regression was implemented using Tensorflow[43] and can be found in https://github.com/biocore/songbird.

**Interpreting ranks.** Supplementary Fig. 2 outlines how to draw hypotheses using the proposed ranking procedure. First the relative differentials need to be computed, preferably using a count-based regression model such as the multinomial regression described above. As noted in the introduction, the coefficients can be represented as centered log ratio (clr) coordinates as follows

$$\beta_k^{(\mathrm{clr})} = \mathrm{clr}(\mathrm{alr}^{-1}(\beta_k)) \tag{14}$$

$$\mathrm{clr}(x) = \left( \log\left(\frac{x_1}{g(x)}\right), \ldots, \log\left(\frac{x_D}{g(x)}\right) \right) \tag{15}$$

where $g(x)$ represents the geometric mean. These coordinates are typically centered around zero, meaning that the chosen reference frame is the center-of-mass, or in other words the average microbe. This is the same reference frame that ALDEx2 and sometimes ANCOM uses. Once the relative differentials are estimated there are two possible analyses. It is possible to construct compositional biplots to visualize all of the regression coefficients and determine how microbes are clustered and driven by metadata covariates. This procedure is outlined in the Songbird tutorial on github.

The other possibility is to identify candidate differentially abundant microbes. To this end, one can construct rank plots (e.g., Figs. 2b and 3a). The rank plots show the ordering of all of the taxa with respect to how much they are associated with a particular metadata covariate, and specific taxa can be highlighted to show their ranks as positions on the rank plot. From the ranks, one can focus on taxa that have very high ranks or very low ranks, since those are the ones that are increasing/decreasing the most relative to each other, and are likely to be important contributors.

These ranks can also help inform which taxa can be used for a suitable reference frame since the difference between the relative differentials can approximate the

effect size that those two microbes will have. As a result, microbes that have very different ranks can be suitable candidates for a log-ratio test. An ideal reference microbe is present across most samples, since this will allow the denominator in a log-ratio to be defined. This was one of the reasons why *Actinomyces* and *P. acnes* were chosen as reference microbes in the case studies. Furthermore, if a microbe is anticipated to be stable across experimental conditions, this could provide additional motivation to select that microbe as the reference microbe.

Zeros will remain to be problematic when comparing log-ratios of taxa among conditions—the procedure used here was to treat zeros as missing data and drop them from the analysis. However, this approach may not be optimal, for instance if two microbes never occur together in the same sample, but one microbe has a very high rank, and the other microbe has a very low rank. The two microbes may have significant explanatory power, but it will not be possible to perform a log-ratio test without imputing the zeros. In scenarios such as this, it may be more appropriate to utilize presence/absence procedures.

While the above procedures provide some recommendations on how to pick an appropriate reference frame, picking a reference frame for hypothesis testing is still an outstanding challenge. Since a reference frame can be defined as the average of a set of microbes, there are $2^N$ possible reference frames for $N$ microbes. Rivera–Pinto proposed one approach towards finding an optimal reference frame[44]; however, this solution maybe suboptimal. Furthermore, it is not clear what properties an optimal reference frame should satisfy, or how false discoveries could be controlled. More theoretical work will need to be done in order to understand statistical properties of these reference frames.

**Simulated benchmarks comparing ANCOM2, ALDEx2, and DR**. We used simulated data to benchmark DR to the output of ALDEx2[45] and ANCOM2[46]. Details can be found in the simulation-benchmarks ipynb at https://github.com/knightlab-analyses/reference-frames. Here, we compared the linear mixed effects model in ANCOM2, the *t*-test in ALDEx2 and ranked multinomial regression coefficients in DR. ALDEx2 determined taxa were significant if the FDR corrected *p*-value fell below 0.05. A taxon was determined to be significant by ANCOM if it passed the 0.9 cutoff.

ALDEx2 and the proposed multinomial regression for DR in this paper use nearly identical models concerning categorical metadata. The major difference is the choice of priors; our model uses a normal prior whereas ALDEx2 uses a Dirichilet prior. As a result, the coefficients from ALDEx2 and the multinomial regression are nearly identical (Supplementary Fig. 1), suggesting that the same ranking procedure can also be applied to the ALDEx2 output. However, ALDEx2 can only handle a single categorical covariate at a time, whereas the multinomial regression proposed can handle multiple covariates, including continuously valued covariates, as shown in the Central Park soils dataset (Fig. 4).

It is important to note that the hypothesis tests that ANCOM and ALDEx2 use may not be consistent with the absolute differentials. Under perfect conditions when the absolute differentials are centered around zero (Supplementary Fig. 1a–d), both ANCOM and ALDEx2 correctly infer that microbes with a differential close to zero are likely not changing.

However, if the center of mass changes and the average microbe is now decreasing on average −2 log fold (Supplementary Fig. 1e–h), both ALDEx2 and ANCOM will incorrectly infer that microbes changing −2 log fold are not changing. In this example, the center of mass reference frame is inappropriate, and leads to predictions that microbes are not changing when they are actually changing on an absolute scale. This highlights difficulties when attempting to link information from relative data to absolute data using hypothesis tests. The hypothesis tests that ALDEx2 and ANCOM perform here are not necessarily incorrect, but could be misleading in situations where microbial load differs dramatically among conditions.

**Interpreting relative differentials through balances**. Balances are ratios of taxa, or groups of taxa, that were previously presented as a valid approach to analyzing compositional data[14]. If we examine the model parameters $\boldsymbol{\beta_k} \in \mathbb{R}^{D-1}$, we reinterpret the quantities given by $\mathrm{alr}^{-1}(\boldsymbol{\beta_k})$ as relative differentials as discussed in Fig. 1. It is also worthwhile to note the connection between $\boldsymbol{\beta_k}$ and balances. Since $\boldsymbol{\beta_k}$ is expressed in alr coordinates, there is also a direct connection to ilr coordinates, meaning that $\boldsymbol{\beta_k}$ can also be transformed into balances. More explicitly, the ilr coordinates of these coefficients can be computed as follows

$$\boldsymbol{\beta_k}^{(\mathrm{ilr})} = \mathrm{ilr}_{\boldsymbol{\Psi}}(\mathrm{alr}^{-1}(\boldsymbol{\beta_k})). \qquad (16)$$

The resulting coefficients are represented as coordinates given by the orthonormal basis $\boldsymbol{\Psi}$. An example of such a basis can be derived from bifurcating trees discussed in Morton et al.[14], Silverman et al.[47], and Washburne et al.[31] This can allow for relative changes in abundances as given by $\mathrm{alr}^{-1}(\boldsymbol{\beta_k})$ to inform which balances are changing in ancestral states given by the tree. The multinomial regression serves as an alternative means to compute regression coefficients discussed in PhILR, Phylofactor and Gneiss, while avoiding issues with imputation and zeros.

**Saliva microbiota study**. Nine volunteers provided unstimulated saliva so that salivary flow rate could be measured according to a standardized protocol[48].

Briefly, individuals were asked to allow saliva to flow for exactly five minutes through a disposable funnel (Simport, SIM F490-2) into a sterile, 15 mL conical tube preloaded with 2 mL sterile glycerol for bacterial preservation. Participants were asked to provide samples before brushing and after brushing teeth in the morning and in the evening. Samples were inverted several times to mix with the glycerol and stored at −20 °C immediately after collection. This study was approved by an Institutional Review Board (IRB# 150275) and written informed consent was acquired before sample collection.

Unstimulated saliva samples were thawed on ice and aliquots were diluted tenfold with sterile, 1x PBS. To remove human cells and salivary debris, samples were filtered using a sterile 5 μm syringe filter (Sartorius Stedim Biotech GmbH). 5 μl 20x SYBR green (SYBR™ Green I Nucleic Acid Gel Stain, Invitrogen) was added to 1 mL of the microbial suspension (0.1x final concentration) and incubated in the dark for 15 min at 37 °C. Finally, 50 μl AccuCount Fluorescent Particles (Spherotech, ACFP-70-10) were added for assessment of microbial load. Samples were processed on a SH800 Cell Sorter (Sony Biotechnology) using a 100 μm chip with the threshold set on FL1 at 0.06%, and gain settings as follows; FSC = 4, BSC = 25%, FL1 = 43%, FL4 = 50%. The gating strategy was adapted from Vandeputte et al.[12] Briefly, fluorescent microbial cells were gated from background on a FL1-Fl4 density plot. Aggregates were excluded by taking the linear fraction on a plot of FL1-height versus FL1-area as previously described[49], and remaining noise was removed by eliminating large events detected on a FSC-BSC density plot (Supplementary Fig. 3). Negative controls (sterile PBS stained identically to samples) were run between each sample set to exclude cross-contamination. Settings were identical among all samples.

DNA extraction and 16S rRNA amplicon sequencing were done using Earth Microbiome Project (EMP) standard protocols (http://www.earthmicrobiome.org/protocols-and-standards/16s). Five hundred microliter of unstimulated saliva was used for gDNA extraction with MagAttract PowerSoil DNA Kit (QIAGEN) as previously described[50]. Amplicon PCR was performed on the V4 region of the 16S rRNA gene using the primer pair 515f to 806r with Golay error-correcting barcodes on the reverse primer. Two hundred forty nanogram of each amplicon was pooled and purified with the MO BIO UltraClean PCR cleanup kit and sequenced on the Illumina MiSeq sequencing platform.

Demultiplexed fastq files were processed using QIIME2 (https://qiime2.org)[51]. Deblur was used to denoise the sequences[52]. Taxonomy was assigned and 16S rRNA gene copy number-corrected using RDP classifier[53] then collapsed to the genus-level. All taxa reported in the manuscript were validated using the NCBI BLAST database[54]. Absolute abundances were estimated by multiplying the total cell-count estimated by flow cytometry by the copy number-corrected microbial proportions from sequencing as outlined above.

For differential abundance testing (Fig. 2e), ALDEx2 determined taxa were significant if the FDR corrected *p*-value fell below 0.05. A taxon was determined to be significant by ANCOM if it passed the 0.6 cutoff. Songbird was used to perform multinomial regression and the repository can be found here: https://github.com/mortonjt/songbird. Paired *t*-tests were performed to evaluate the differences before and after brushing teeth. All log-ratios that were evaluated to either positive or negative infinity were dropped prior to statistical analysis.

**Analyzing the Central Park soils study with reference frames**. Data from Ramirez et al.[30] were retrieved from Qiita[55] (https://qiita.ucsd.edu/study/description/2104). Amplicon sequence variants that appeared in less than 24 samples were filtered out, reducing the number of analyzed taxa to 30, 248 taxa. This filtering criteria was chosen to ensure that each of the six covariates had at least four samples to fit against. The following multinomial linear model was estimated

$$y_i \sim \mathrm{Multinomial}(x_i \beta)$$
$$\beta = [\beta_0, \beta_{\mathrm{water}}, \beta_{\mathrm{nitro}}, \beta_{\mathrm{pH}}, \beta_{\mathrm{carbon}}, \beta_{\mathrm{biomass}}]$$

where $\boldsymbol{y_i}$ represents the microbial relative abundances in sample $i$, $x_i$ are the measured covariates for sample $i$, $\beta_{\mathrm{water}}$ are the relative differentials with regard to water content, $\beta_{\mathrm{nitro}}$ are the relative differentials with regard to nitrogen concentration, $\beta_{\mathrm{pH}}$ are the relative differentials with regard to pH, $\beta_{\mathrm{carbon}}$ are the relative differentials with regards to carbon measurements and $\beta_{\mathrm{biomass}}$ are the relative differentials with regards to measured biomass.

**Shotgun metagenome studies**. We used supplementary data from Byrd et al.[27] and Leung et al.[28]. The provided relative abundances were compared to log-ratios of given taxa from the raw count data. Paired *t*-tests were performed to evaluate the differences between lesion and non-lesion skin samples. All log-ratios that were evaluated to either positive or negative infinity were dropped prior to statistical analysis. These numerical issues occur when particular microbes are not observed, and we treat them as missing data, respectively.

**Reporting summary**. Further information on research design is available in the Nature Research Reporting Summary linked to this article.

## Data availability

The sequences and biom tables[56] from the saliva microbiota study can be found on Qiita (http://qiita.microbio.me)[55] under study ID 11896 and at EBI under ERP111447.

## Code availability

All analyses can be found under https://github.com/knightlab-analyses/reference-frames

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

## Acknowledgements

We would like to thank Huang Lin and Shyamal Peddada for running ANCOM2 on the oral microbiome study and providing the R scripts. We'd also like to acknowledge Cara Magnabosco for her insights on soil microbes in the Central Park soils study and Doris Vandeputte for her feedback on the absolute quantification normalization. J.T.M. was funded by NSF grant GRFP DGE-1144086. C.M. was funded by NIDCR F31 Fellowship 1F31DE028478. This work was funded in part by a Seed grant from the Center of Microbiome Innovation, UC San Diego.

## Author contributions

J.T.M. led the data analysis and developed the idea of ranking differentials. C.M. collected saliva samples, performed flow cytometry and sequencing, and provided biological context behind oral microbial biofilms and atopic dermatitis. A.W. contributed the idea of reference frames. J.S. contributed the concept of bias due to total biomass differences. L.S.Z. processed the shotgun data for atopic dermatitis and provided background knowledge for atopic dermatitis. A.E. provided background knowledge behind oral microbial biofilms. K.Z. and R.K. provided insights on biological validation. All authors wrote and proofread the manuscript.

## Additional information

**Competing interests:** The authors declare no competing interests.

