## [Peer Review File · Nature Communications]

REVIEWERS' COMMENTS:

Reviewer #2 (Remarks to the Author):

The merit of this work is that it brings the compositional nature and its dangers under the attention of scientific community. However, issues related to the compositionality of microbiome data have been reported already many times (as is correctly mentioned in the manuscript). Some solutions have been proposed, but I must admit that there is still much room for improvement, particular with respect to the interpretation of results based on log-ratios and problems related to the many zero counts. The authors first demonstrate several issues related to the compositional nature and then continue with advocating the use of ratios. First the ratio of counts of taxa pairs is discussed, followed by a novel approach based on ratios of counts of the same taxon in two conditions (e.g. before - after treatment, or two treatment groups). They show that the ranking of these ratios is invariant to the total microbial load, making it applicable to relative abundance 16D RNA sequencing data. They continue with a framework for selecting a “reference frame”, relative to which the ratios can be computed. Their approach, and its added value, is illustrated on a few datasets. They conclude that many published microbiome studies should be re-analysed with their method, because (1) wrong conclusions may have been published, and (2) new results may be reached with their method.

The current version is a great improvement over the original manuscript.

Comments:

1. p. 7., starting from l. 113 Here it starts to get confusing, because now ratios may also refer to ratios between different taxa, whereas on the previous page the ratios refer to ratios between conditions. Can the difference be made more explicit?
2. p. 19, l. 332. I think it must be $<$ instead of $>$.
3. p. 7, l. 328 and 332: why “at least on i ”?

4. p. 20. It would be helpful do define alr upfront.
5. p. 21, l. 356. It would be helpful to explain that $g(x)$ is the geometric mean.
6. It is still not entirely clear to me how the ranks of the relative differentials are used for selecting a reference frame. I think that the authors try to give guidelines on p. 22, lines 369-376, but I personally find this far from clear how to use these guidelines in practice. Also the case studies presented earlier in the manuscript do help me very much. As far as I understand, in all case studies the reference frame was one taxon. It was selected as a taxon with a large relative differential, but not necessarily the largest(?).
On p. 22, lines 369-376, the authors suggest to apply statistical tests for comparing taxa with very different DR. Does this imply that the authors suggest not to test all taxa, but only those with very different DR? This looks like data-driven hypotheses, and then the risk for false positive results increases in an uncontrolled manner.
I suggest to make the procedure for selecting the reference frame very explicit.
7. p. 22. The authors mention the log ratio test. Do they mean a t-test applied to the log ratios of the counts?
8. p. 24. It would be helpful to remind the reader, who is not familiar with the mathematics behind compositional data, about the definition of balances.
9. Figure 2. In the caption the explanations of panels (b) and (c) should be switched.
10. Fig 2(c). The labels of the axes are confusing because the terms “raw t -statistics“ and “cell t -statistics” are not introduced.
11. Suggestion: maybe it is helpful to show the reader the following equality:

$$\log \frac{a_i/N_a}{b_i/N_b} - \log \frac{a_j/N_a}{b_j/N_b} = \log \frac{a_i/b_i}{a_j/b_j} = \log \frac{a_i/a_j}{b_i/b_j}$$

Reviewer #3 (Remarks to the Author):

The authors addressed the issues I identified during the initial review. I have no remaining issues and recommend the manuscript for publication in Nature Communications.

Point-by-point Response to Reviewers

Reviewer #1

No comments

Reviewer #2

The merit of this work is that it brings the compositional nature and its dangers under the attention of scientific community. However, issues related to the compositionality of microbiome data have been reported already many times (as is correctly mentioned in the manuscript). Some solutions have been proposed, but I must admit that there is still much room for improvement, particular with respect to the interpretation of results based on log-ratios and problems related to the many zero counts. The authors first demonstrate several issues related to the compositional nature and then continue with advocating the use of ratios. First the ratio of counts of taxa pairs is discussed, followed by a novel approach based on ratios of counts of the same taxon in two conditions (e.g. before - after treatment, or two treatment groups). They show that the ranking of these ratios is invariant to the total microbial load, making it applicable to relative abundance 16D RNA sequencing data. They continue with a framework for selecting a “reference frame”, relative to which the ratios can be computed. Their approach, and its added value, is illustrated on a few datasets. They conclude that many published microbiome studies should be re-analysed with their method, because (1) wrong conclusions may have been published, and (2) new results may be reached with their method.

The current version is a great improvement over the original manuscript.

Thank you for this succinct summary and we are very grateful for your constructive feedback.

Comments:

1. p. 7., starting from l. 113 Here it starts to get confusing, because now ratios may also refer to ratios between different taxa, whereas on the previous page the ratios refer to ratios between conditions. Can the difference be made more explicit?

Thank you for bringing up this potential source of confusion. In the previous section we define the term ‘differential’ as being the logarithm of the fold change in abundance of a taxa between two conditions. As we noted in the manuscript, there is a bias due to the change of the microbial-load, so only the ranks of this differential can be reliable. This is different from the log-ratios that are computed on a per-sample basis - but there are connections between the log-ratios and the ranks as you pointed out in your comments.

2. p. 19, l. 332. I think it must be < instead of >.

Yes! Thank you for catching this typo.

3. p. 7, l. 328 and 332: why “at least on i”?

This actually applies to all i - thank you for catching this.

4. p. 20. It would be helpful do define alr upfront.

Thank you for this suggestion. We have updated the sentence preceding the equation to read 'Counts from the multinomial regression can be formulated using additive log-ratio transformation (alr) in the following generative model'

5. p. 21, l. 356. It would be helpful to explain that $g(x)$ is the geometric mean.

Thank you for this suggestion. We have updated the sentence following this equation to read 'where $g(x)$ represents the geometric mean'.

6. It is still not entirely clear to me how the ranks of the relative differentials are used for selecting a reference frame. I think that the authors try to give guidelines on p. 22, lines 369-376, but I personally find this far from clear how to use these guidelines in practice. Also the case studies presented earlier in the manuscript do help me very much. As far as I understand, in all case studies the reference frame was one taxon. It was selected as a taxon with a large relative differential, but not necessarily the largest(?).

Choosing relevant reference frames remains to be a challenging task. This is especially true due to the nature of microbiome sequencing data, namely sparsity and taxonomy limitations. For N microbes there are 2^N possible reference frames one could choose (since one could take the average of a set of microbes as a reference as well) - these are the approaches that have been utilized with the balances, namely Philr, Phylofactor and Gneiss. We have presented a simplified approach where individual microbes can serve as references and all possible reference frames containing a single microbe can be ranked and visualized on a single plot.

On p. 22, lines 369-376, the authors suggest to apply statistical tests for comparing taxa with very different DR. Does this imply that the authors suggest not to test all taxa, but only those with very different DR? This looks like data-driven hypotheses, and then the risk for false positive results increases in an uncontrolled manner.

We don't aim to downplay the role of FDR correction - more work will need to be done on how to properly access this. The goal of this work is to highlight the counter-intuitive implications behind differential abundance, which would invalidate existing FDR correction efforts if the appropriate null hypothesis is not correctly realized. We have added a couple of sentences at lines 392 - 394, thank you for pointing this out.

I suggest to make the procedure for selecting the reference frame very explicit.

Defining an algorithm to find the best reference frame is outside the scope of this paper, since there are many ways to do this - we have cited a few ways to do this in the manuscript. The reference frames that we chose in the paper were chosen provide biological intuition to motivate why identifying reference frames is important. However, the reference frames that we chose may not be optimal. We outlined the problem more explicitly in the methods section on line 368.

7. p. 22. The authors mention the log ratio test. Do they mean a t-test applied to the log ratios of the counts?

Thank you for highlighting this confusing sentence. We meant comparing log-ratios among conditions and have updated this sentence to read: "Zeros will remain to be problematic when comparing log-ratios of taxa among conditions - the procedure used here was to treat zeros as missing data and drop them from the analysis."

8. p. 24. It would be helpful to remind the reader, who is not familiar with the mathematics behind compositional data, about the definition of balances.

Thank you for this helpful suggestion. We have added the following sentence to the beginning of this section: 'Balances are ratios of taxa, or groups of taxa, that were previously presented as a valid approach to analyzing compositional data (Morton et al., 2017)'

9. Figure 2. In the caption the explanations of panels (b) and (c) should be switched.

Thank you for catching this! We have corrected the legends accordingly.

10. Fig 2(c). The labels of the axes are confusing because the terms “raw t-statistics“ and “cell t-statistics” are not introduced.

We agree, thank you for pointing this out. We have updated the y-axis to read ‘abs. T-test’ and ‘rel. T-test’ and have defined these terms in the text.

11. Suggestion: maybe it is helpful to show the reader the following equality:

$$\log \frac{a_i/N_a}{b_i/N_b} - \log \frac{a_j/N_a}{b_j/N_b} = \log \frac{a_i/b_i}{a_j/b_j} = \log \frac{a_i/a_j}{b_i/b_j}$$

Thank you for this comment, especially given that it yields a clear connection between the log-ratios and the ranks. This expression has been added to the online Methods section.

Reviewer #3 (Remarks to the Author):

The authors addressed the issues I identified during the initial review. I have no remaining issues and recommend the manuscript for publication in Nature Communications.

Thank you for your helpful comments. They have greatly improved this manuscript!